# A Review of Medical Conditions and Behavioral Problems in Dogs and Cats

**DOI:** 10.3390/ani9121133

**Published:** 2019-12-12

**Authors:** Tomàs Camps, Marta Amat, Xavier Manteca

**Affiliations:** 1Etovets: Behavioral Medicine and Animal Welfare, 07010 Palma, Spain; 2School of Veterinary Medicine, Universitat Autònoma de Barcelona, 08193 Bellaterra, Spain; marta.amat@uab.es (M.A.); xavier.manteca@uab.es (X.M.)

**Keywords:** behavioral problems, medical conditions, hypothyroidism, sleep disorders, neurological problems

## Abstract

**Simple Summary:**

Behavioral problems and medical conditions have been treated separately for years. However, behavior depends directly on an animal’s health condition, and vice versa. Some behavioral problems are caused totally or partially by a medical condition. Additionally, some of these problems represent a diagnostic challenge for veterinarians because, in many cases, apart from behavioral changes, there are no other clinical signs or evidence of illness. Improving our knowledge of the most common medical problems that can modify behavior may help veterinarians to improve their diagnostic protocols and treatments. Based on our experience, most relevant medical conditions are some neurological problems, endocrine or metabolic problems, and pain-related conditions. Thus, the aims of this review are to describe the state of the art of the relationship between these medical conditions (among others) and behavioral problems, and proposing new lines of investigation.

**Abstract:**

Not all animals behave identically when faced with the same situation. These individual differences in the expression of their behavior could be due to many factors, including medical conditions. These medical problems can change behavior directly or indirectly. The aims of this review are to describe the state of the art of the relationship among some medical and behavioral problems, and to propose new lines of investigation. The revision is focused on the relation between behavioral problems and pain, endocrine diseases, neurological problems, vomeronasal organ alterations, and cardiac disorders. These problems represent a diagnostic challenge from a practical point of view. The most common sign of pain in animals is a change in behavior. Although the relation of pain to behavioral problems has been widely studied, it is not absolutely clear. As an example, the relation between sleep disorders and pain is poorly known in veterinary medicine. New studies in humans and laboratory animals show that a reciprocal relationship does, in fact, exist. More specifically, the literature suggests that the temporal effect of sleep deprivation on pain may be stronger than that of pain on sleep. Some behavioral problems could modify the sleep–awake cycle (e.g., cognitive dysfunction). The impact of these behavioral problems on pain perception is completely unknown in dogs and cats. Thyroid hormones play an important role, regarding behavioral control. Both hypothyroidism and hyperthyroidism have been related to behavioral changes. Concerning hypothyroidism, this relationship remains controversial. Nonetheless, new neuro-imaging studies provide objective evidence that brain structure and function are altered in hypothyroid patients, both in laboratory animals and in humans. There are many neurological problems that could potentially change behavior. This paper reviews those neurological problems that could lead to behavioral changes without modifying neurological examination. The most common problems are tumors that affect central nervous system silent zones, mild traumatic brain injury, ischemic attacks, and epilepsy. Most of these diseases and their relationship to behavior are poorly studied in dogs and cats. To better understand the pathophysiology of all of these problems, and their relation to behavioral problems, may change the diagnostic protocol of behavioral problems.

## 1. Introduction

Not all animals behave identically when faced with the same situation. These individual differences in the expression of their behavior could be due to many factors. Genetics, pre-natal manipulation of the dam [1,2], experiences of the animal during the different developmental stages (especially important, neonatal and socialization periods) [3], experiences during adulthood, and the correct functioning of organs and systems are among these factors.

Medical conditions can modify the proper functioning of organs and systems. Thus, these medical problems can change behavior directly or indirectly. These problems can be divided into four main groups: (1) Problems that modify or prevent the perception of the environment (e.g., blindness), (2) problems that change the processing of the perceived information (e.g., intracranial tumor) or alter the internal processes (hormonal and/or neurological) involved in behavior (e.g., hypothyroidism), (3) problems that induce a stress response that can modify behavior (e.g., pain), and (4) problems that change or prevent the expression of behavior (e.g., a broken leg).

Behavioral problems can be defined in different ways. However, all definitions have at least one or more of the following features. First, behavioral problems are behaviors that must be perceived as annoying by the owners. This fact confers an important degree of subjectivity to the definition and, what is more important, to the recognition of the problem by the owners. Second, they can be dangerous for people, other animals, or even for the patient itself. Third, most behavioral problems, directly or indirectly, can deteriorate the patient’s welfare. Finally, they can appear as a consequence of a medical condition or pathology (i.e., repetitive behavior due to an intracranial tumor) but, also, they can be absolutely normal behaviors that are considered as problematic by the owners (i.e., territorial aggression, competitive aggression, etc.). There are also medical problems that can make a behavioral problem worse (i.e., pain and competitive aggression or partial blindness and fears or phobias). Although the relationship between medical and behavioral problems has been widely investigated, many doubts still exist in that regard. For example, could medical conditions lead to behavioral problems with a “normal and coherent” behavior pattern? Are all behavioral problems, caused by medical conditions, easy to recognize only with the clinical or behavioral history of the animal? Are all of these problems expressed as a change in an animal’s behavior during adulthood, or can the pathology be present from childhood, even when behavioral signs appear when the animal is an adult? All of these questions, among others, are important in order to improve the prevention, diagnostic, and treatment protocols in behavioral medicine.

The aims of this review are to describe the state of the art of the relationship between some medical and behavioral problems, and proposing new lines of investigation.

## 2. Material and Methods

Problems included in this revision are those that can contribute to behavioral problems, but which are difficult to differentiate from a true behavioral or medical problem. We have included those problems that represent a diagnostic challenge from a practical point of view.

The first review has been performed using Pubmed and Google Scholar. References cited in the articles found in the review have also been taken into account.

## 3. Pain and Behavioral Problems

The most common sign of pain in animals is a change in behavior [4]. Behavioral signs of pain include both the loss of normal behaviors and the development of new and abnormal behaviors. The most common behaviors classified as “lost normal behaviors” are decreased ambulation or activity, decreased interaction with other pets or family members, lethargic attitude, decreased appetite, and decreased resting behaviors. The most common “developed abnormal behaviors” are aggression, fear reactions, inappropriate elimination, vocalization, altered facial expression, altered posture, restlessness, and hiding [4]. Pain has also been related to repetitive behaviors [5].

From a clinical point of view, pain should be included in the differential diagnoses of aggressive behaviors, fear and phobia reactions, sleep disorders, inappropriate elimination, vocalization, and repetitive behaviors. However, according to the authors’ experience, at least in referral services, pain-related vocalizations are not reported very often as a behavioral problem by the owners.

### 3.1. Pain and Aggressive Behaviors

It is widely accepted that pain can lead to aggressive behaviors that are often described as a defensive reaction to avoid physical contact that may cause further injury [6]. Moreover, the animal can learn from a painful experience [7]. If it has experienced pain in a specific context, it may avoid the same or a similar context in the future, displaying passive behaviors (e.g., fear and phobias) or more reactive ones (e.g., fear-related aggressive behaviors), even when the animal is not yet in pain. 

Nevertheless, it is also well-described that pain elicits a stress response [8]. Pain-induced responses lead to many physiological and behavioral changes. Decreased serotonin activity in the brain would be among them [8]. Moreover, pain could also decrease physical activity, and this may further reduce serotonin activity in the central nervous system (CNS) [9]. Finally, a reduction of serotonin activity in the CNS has been related to aggressive behaviors [10].

It could be that these two different mechanisms may result in a different pattern of aggressive behavior caused by pain. One study based on 12 clinical cases showed that two different patterns of pain-related aggression exist, depending on whether the dog was aggressive before the onset of pain [11]. According to this clinical case study, dogs that were aggressive before the onset of pain may be more aggressive (more frequent and more intense) in the same context of their previous aggression and tend to be less impulsive. Otherwise, dogs that were not aggressive before the onset of pain may be more impulsive, and the aggression may happen in a context of manipulation. In other words, this second group seems to display avoidance behaviors, whereas the first group may decrease the aggression threshold as a response to pain (Appendix A, CASE 1). Finally, this second group is a diagnostic challenge, which means that all patients, including those with a “normal and contextual aggression pattern,” should be closely checked in order to rule out an underlying pain condition.

### 3.2. Pain and Fears

Pain can also induce fear. First, pain acts as an unconditional stimulus, which induces a fear response [12,13]. Therefore, when an animal is in a situation in which it experiences pain, it will try to create associations between the stimulus that causes pain and other neutral stimuli [7,14]. These neutral stimuli may help to predict a similar situation in the future. When the conditional context is present again, the animal may show fear even in the absence of the initial unconditional stimulus.

A second mechanism exists that is demonstrated in humans, but not in animals. It is well-known that people and animals that are in pain, especially a chronic one, are more likely to suffer from anxiety [5,15]. Due to this anxiety state, these people have a higher probability of showing a pessimistic cognitive bias, which makes initially neutral stimuli become potential (unreal) sources of pain, which may lead to more anxiety and fear [16]. However, a similar process could also happen/occur in animals. A clinical case report of two dogs with fear towards unconventional stimuli was presented [17]. These dogs did not respond to conventional treatment of fear. After months, the animals were also diagnosed with pain. Subsequently, the dogs were treated simultaneously for fear and pain, and the problem was solved.

It is well-known that pain can also lead to anxiety in dogs [18]. Moreover, dogs with another source of anxiety (separation anxiety) exhibit a pessimistic cognitive bias [19]. It may be that animals with pain can suffer from anxiety, and that anxiety leads to a pessimistic cognitive bias that also leads to fear reactions in dogs. Although further studies are needed in order to confirm this hypothesis, one recent clinical case study [20], with a small sample size, suggests a qualitative relationship between pain and noise sensitivity. In this study, 20 dogs with noise sensitivity were divided in two groups, 10 dogs that showed noise sensitivity and pain (clinical case group), and 10 that only showed noise sensitivity (control group). In the “clinical cases” group, the age of onset of the noise sensitivity was, on average, 4 years later than “control cases”. Additionally, dogs with pain were more likely to generalize their fear to contextual (environmental) clues and toward other dogs. Finally, “clinical cases” responded well to treatment once the involvement of pain had been identified.

### 3.3. Pain and Sleep Disorders

It is widely accepted that pain and sleep are related. However, many questions remain concerning the direction of causality. Brain structures associated with the generation and maintenance of sleep are also involved in pain modulation, providing a neurobiological substrate for a reciprocal relationship [21]. Early longitudinal and experimental evidence reviewed by many authors in the human literature revealed that a reciprocal relationship exists [21,22,23], where pain results in sleep disturbance and disturbed sleep enhances pain. It seems that acute pain can lead to a night of poor sleep in a human patient who had no previous sleep complaints, with effects that are usually reversible [24]. In the case of chronic pain, this relationship is more controversial. In fact, a trend in the literature suggests that the temporal effect of sleep deprivation on pain may be stronger than that of pain on sleep [23]. In other words, insomnia symptoms significantly increase the risk of developing future chronic pain disorders in previously pain-free individuals, whereas existing pain is not a strong predictor of new incident cases of sleep disturbances. Additionally, good sleep increases the chance that chronic pain will improve over time.

There is a lack of studies in dogs and cats regarding this issue. Nevertheless, it could be interesting to investigate it. First, it may be a good model for human medicine. Secondly, behavioral problems in small animals that lead to sleep alterations are common, above all, in elderly patients (cognitive dysfunction syndrome) [25]. It may be interesting to investigate how sleep alterations can modify the pain threshold in those animals in order to guarantee their welfare. Finally, it could be interesting to investigate how sleep deprivation, at hospitals or veterinary centers, can change the perception of pain and the use of analgesic drugs when dogs and cats are hospitalized, especially because it is common that the sleep–wake cycle of these patients is not respected.

### 3.4. Inappropriate Elimination

As has been mentioned before, pain acts as an unconditional stimulus, which induces a fear response [7,12]. Feline urinary tract disease is common in cats and leads to pain in most cases [26]. Therefore, if a cat is in pain when urinating, it can cause a relation between pain and the litter tray. In other words, pain is a common cause of litter tray aversion in cats. A similar process may happen with feces and problems that induce pain during defecation [27,28]. Cats with osteoarthritis, especially in hind limbs, can eliminate out of the litter tray because of pain. These cats tend to avoid bending when urinating and, even when they are standing in the litter tray for eliminating, the owner can find these depositions on the floor, out of the tray. Analgesic drugs and big litter trays could help to solve this kind of problem.

### 3.5. Repetitive Behaviors

Pain and compulsive disorders are related. In humans, individuals who suffer from chronic pain show more anxiety disorders than the general population (35% against 17%) [5]. Anxiety disorders can induce obsessive–compulsive disorders, among others. As an example, women with fibromyalgia are 4–5 times more likely to have had a lifetime diagnosis of obsessive–compulsive disorder, post-traumatic stress syndrome, or generalized anxiety [29].

There is a lack of specific studies about the real role of pain on compulsive disorders in animals. However, two mechanisms could be involved. First, it is widely accepted that stress plays an important role in the development of compulsive behaviors in animals, as well as in humans. Pain elicits a stress response in animals [8], and could act as a trigger of a compulsive behavior by itself, as a source of stress. Secondly, pain may lead to a licking behavior directed toward the painful area of the body. Thus, this licking behavior could be reinforced by itself because it decreases the sensation of pain [30], and also by the owners because they often pay attention to the animal when it performs the behavior (operant conditioning). A good example could be the licking behavior directed toward the caudal area of the abdomen in cats with bladder pain.

## 4. Endocrine Diseases and Behavioral Problems

Thyroid hormones play an important role regarding behavioral control. In fact, the brain is a major target organ for thyroid hormones [31]. Thyroid hormone alterations have been associated with behavioral problems in animals, as well as humans. Nonetheless, some authors think that causality of the relationship is unlikely and defend that the problems may be just co-morbid and co-exist in time [32]. Nevertheless, some new data strongly suggest that a causative relationship could exist. New imaging studies provide objective evidence that brain structure and function are altered in hypothyroid patients, with decreased hippocampal volume, cerebral blood flow, and function globally, and in regions that mediate attention, working memory, and motor speed in humans and in rats [33,34,35,36,37]. In addition, a recent study showed that some of these alterations (alterations in working memory and abnormalities in functional magnetic resonance) were no longer present after six months of treatment with levothyroxine [36]. Finally, thyroid hormones have been found to affect the turnover of serotonin [38,39]. Serotonin is involved in the control of behavior (e.g., aggression and fear) [40].

Similar studies in dogs do not exist. However, in clinical reports, hypothyroidism has been classically associated with aggression, and also with apathy, lethargy or mental dullness, cold intolerance, exercise intolerance, and decreased libido [41]. Additionally, treatment with levothyroxine improves clinical signs, including aggressive behaviors [42]. Nevertheless, some authors argue that levothyroxine can change behavior by itself, including normal and euthyroid animals [43]. For this reason, and in the absence of double-blind control studies, the link remains speculative for some authors [32,43].

In human medicine, as a summary, overt hypothyroidism (elevated serum thyroid-stimulating hormone (TSH), and low, free-thyroxin level) is associated with clinically significant neuropsychiatric decrements that are largely reversible with levothyroxine treatment, whereas mild or subclinical (elevated TSH with normal, free thyroxin) is not associated with severe or widespread neuropsychiatric decreases [31]. In this case, if neuropsychiatric symptoms are present, they are usually unrelated to the thyroid problem, and these do not reliably reverse with thyroid supplementation. Veterinary medicine has a similar approach. In all dogs where a behavioral problem co-exists (especially important aggressive and fear-related behaviors) with an overt hypothyroidism, the supplementation is recommended and often solves or improves the behavioral problem. However, in euthyroid dogs (having low total T4 and/or free T4, with TSH within the normal values and other pathologies or with concomitant use of other drugs), treatment with levothyroxine is not recommended, in spite of having a behavioral problem and low serum concentration of T4 [32]. In these cases, it is unlikely that the behavioral problem and the thyroid alterations are related.

In dogs, all cases of aggressive-related problems should be checked for hypothyroidism because of the fact that, in some cases, the pattern of aggression is absolutely normal and contextualized (for example, a dog is only aggressive in a competitive context) [42]. So, the only way to rule out hypothyroidism as the cause of any aggression in dogs is blood testing.

Hyperthyroidism has been associated with behavioral changes in animals and in humans. It is well-known that hyperthyroidism has a major effect on the development of the nervous system [44,45,46,47,48,49,50,51,52]. Hyperthyroidism during developmental phases may result in an irreversible impairment, morphological abnormalities, disorganization, or cytoarchitectural abnormalities [51], and these changes would be permanent. Nevertheless, from a clinical point of view, and although a causative relationship has not been proven, a relationship between hyperthyroidism in cats and aggression has been suggested [43]. Hyperthyroidism in cats is not typically a developmental disease. In fact, it is a disease of geriatric cats with a mean age of 13 years [53]. Nevertheless, it is known that intraperitoneal administration of T3 and T4 (for seven days) increased the number of cortical beta-adrenergic and serotoninergic receptors in adult rats [38]. Additionally, recent data suggest that iatrogenic hyperthyroidism in adult rats also changes the levels of dopamine, norepinephrine, and serotonin in different brain regions, as well as in blood plasma, cardiac muscle, and the adrenal glands [54]. All of these neurotransmitters play an important role in behavioral control. Thus, this relationship may explain, at least in part, behavioral changes seen in adult and elderly animals with hyperthyroidism. On the other hand, other authors suggest that this relationship occurs only as a co-morbid condition [32].

## 5. Neurology and Behavioral Problems 

The central nervous system (CNS) is directly involved in the control of behavior. Thus, there are many neurological problems that could potentially change behavior. The behavioral changes due to neurological issues could be divided into four subgroups (Table 1). First, animals that show behavioral changes, changes in the neurological examination, and changes in the laboratory and/or imaging work-up. Second, animals that have behavioral changes and changes in the neurological examination, but without having changes in the work-up. Third, animals that have behavioral changes without having neurological changes, but show work-up alterations. Finally, there are animals that show behavioral changes due to neurological alterations, but without changing neurological and laboratory or imaging work-up.

Problems of Groups 1 and 2 are not further discussed here, because they are problems easy to differentiate from behavioral problems and are commonly diagnosed and treated by neurologists. However, we need to stress two important points. First, from our experience, we can suggest that liver malfunctioning (i.e., portosystemic shunt) can lead to behavioral problems that are very difficult to recognize in dogs. In some cases, behavioral changes are the only clinical signs in these dogs, especially anxiety- and/or fear-related behaviors. Nevertheless, in most cases, these clinical signs appear in the adults and without a consistent context (Appendix A, CASE 2). Second, we have to stress the role of behaviorists in the treatment of animals with behavioral alterations caused by neurological problems. For example, in human medicine, when an intracranial tumor is removed, neuropsychologists and/or neuropsychiatrists are involved in the treatment and collaborate with neurologists. In other words, in veterinary medicine, when a patient is presented because of behavioral problems that are caused by a neurological problem, this does not exclude the important role of the behaviorist in the treatment.

There are areas in the CNS that are silent to the neurological examination [55]. In humans, frontal and prefrontal cortexes are two of those areas. Frontal lobes have been identified experimentally as silent zones in animals [56]. Some of these areas have a major role in behavioral control. Animals that have lesions (e.g., tumors, ischemic lesions, etc.) in those areas could show behavioral changes (and/or seizures) without having changes in neurological examinations. One retrospective study that included 43 dogs with tumors that affected the rostral brain showed that 22 dogs had seizures alone as their initial sign, four dogs had seizures and behavioral changes upon initial examination, and five dogs had abnormal behavior patterns only [55]. Additionally, 31 of the 43 dogs had a normal neurological examination upon initial presentation. However, 25 of these 31 dogs later developed persistent neurological deficits. Importantly, eight dogs never developed neurological deficits and were euthanized because of uncontrolled seizures or unacceptable behavior. It is important to realize that in this study, the age of the dogs ranged from 5 to 15 years, which does not overlap with the mean age of onset of behavioral problems (≈12–24 months of age) [32]. Thus, in practical terms, the age of onset of the problem could be used in these animals as an indicator that more invasive laboratory and imaging work-up is justified.

Finally, some neurological problems change behavior without changing neurological examination or imaging or laboratory tests. All of these problems have been described in humans, and some of them also in laboratory and companion animals. The most common problems in this group are: Idiopathic epilepsy, mild traumatic brain injury, and transient ischemic attacks.

Seizures are the most common neurological problem reported in dogs that are owned as pets [57]. Seizures result from abnormal electrical discharges in the brain. The most commonly given diagnosis for canine seizures is idiopathic epilepsy [58]. There are different classifications in the veterinary literature for seizures. Nonetheless, empirical studies have systematically classified canine seizures as generalized and partial or focal [59]. In fact, partial seizure with secondary generalization seems to be the seizure type most observed in dogs [57,60,61]. Generalized as well as partial seizures have been related to behavioral changes in dogs and cats.

Generalized seizures are caused by abnormal electrical discharges in large areas of the brain. In humans, several and an increasing number of studies have identified a bi-directional relationship between psychiatric disorders and recurrent seizure disorders [62,63,64,65,66]. This co-morbidity has also been reported in laboratory rats [67,68] and companion dogs [69,70]. Depression and anxiety disorders, followed by psychoses and attention deficit disorders, are the most common psychiatric disorders in human medicine. Similarly, Shihab and colleagues showed that dogs with epilepsy have a higher risk of displaying fear/anxiety-type behaviors and defensive aggression and show abnormal perception (that includes barking without apparent cause, chasing shadows or lights, aimless pacing, and staring into space). Other studies also have established a relationship between idiopathic epilepsy and cognitive impairment in dogs [71,72]. Interestingly, this relationship seems to be bi-directional. People with a history of depression or suicide could have up to a 7-times greater risk of developing epilepsy [66,73]. Finally, one review study suggests [74], as occurs in humans, that behavioral modification plans that aim to reduce anxiety in dogs can help epileptic dogs to reduce the number of episodes and improve their quality of life.

On the other hand, in partial seizures, caused by localized abnormal discharge, the clinical signs depend on the affected area of the brain. Thus, partial seizures have been associated with different behavioral alterations, both in humans and animals. The human ILAE (International League Against Epilepsy) classification of partial seizures has been used as a model of classification of animal partial seizures [75]. Berendt et al. studied 70 dogs [75] with a confirmed diagnosis of epilepsy with partial seizures with or without secondary generalization. They recorded the signs of partial seizure activity following the human ILAE classification, and found that 80% of dogs (*n* = 56) showed paroxysms of behavior, 69% of dogs (*n* = 48) showed motor signs, and autonomic signs were recorded in 23% of dogs (*n* = 16). Importantly, nine dogs (13%) had partial seizures without secondary generalization. The majority of dogs showed a combination of signs of, at least, two groups of signs. However, a pool of dogs could exist with behavioral changes due to partial seizures without any other evident neurological sign seen by the owner. Additionally, these dogs might not show any abnormality during the neurological examination. Finally, in 48 dogs (69%), the first seizure was observed before or at 3 years of age [75], which overlaps with the mean age of onset of behavioral problems (≅12–24 months) [32]. All of these highlight the difficulty of the diagnosis and treatment of those animals, above all, considering that some drugs used in behavioral medicine could decrease the epileptic threshold [76].

Traumatic injury is a common occurrence in veterinary medicine [77], and traumatic brain injury occurs in a high proportion of these animals [78]. A single-center retrospective study reported that up to 25% of dogs with severe blunt-force trauma suffer from traumatic brain injury (TBI) [77]. TBI is defined as structural injury or physiological disruption of the brain induced by an external force [79]. Although dogs with TBI are assessed clinically using the modified Glasgow coma scale (GCS), there is no standardized classification of severity for TBI in animals. There is an index of disease severity called improved survival prediction index (SPI2), but it is not specific for trauma patients, and it relies on the most severe values for the first 24-h period after admission to the ICU service [80]. Nevertheless, in human medicine, TBI is classified into three different categories of severity: Mild, moderate, and severe, depending on GCS, along with four more values (structural imaging, loss of consciousness, alteration of consciousness/mental state, and post-traumatic amnesia).

There are studies in human medicine with mild TBI (mTBI) that highlight changes in conventional clinical neuro-imaging (MRI scanning or computed tomography (CT)). However, these changes are present in a low percentage of people. Only 5–10% of people with mild TBI showed abnormalities in CT [81,82,83,84]. Nevertheless, thanks to studies in humans and experimental animal models, it is well-known that mTBI leads to diffuse axonal injury because of biochemical forces and a host of injury-mediated cytotoxic processes [85]. Additionally, acute and chronic alterations of neurotransmitter production and/or delivery have been associated with mTBI [86,87,88]. Most of these alterations are visible using functional neuro-imaging techniques. They more accurately reflect the extent of damaged tissue than either conventional CT or MRI [85]. Studies with both single photon computed tomography (SPECT) and positron emission tomography (PET) suggest that TBI leads to disturbances in brain function, even when there are no abnormalities in conventional neuro-imaging techniques (CT and MRI) [89,90]. The use of these techniques in veterinary medicine as a diagnostic tool is anecdotal.

Importantly, the vast majority of people fully recover during the first year following mTBI [85]. However, from 1 to 20% of people who have suffered from TBI will develop persistent cognitive, emotional, behavioral, and/or physical impairments that last for more than one year following the TBI. The most common cognitive, emotional, or behavioral changes are irritability, depression, apathy, impulsiveness, and attention and/or memory impairments [85]. There is a lack of studies in veterinary medicine about the impact of TBI on behavior. Nevertheless, it could be interesting to study this event in veterinary medicine due to two reasons. First, as we previously mentioned, because traumatic injury is common in veterinary medicine [77], and TBI occurs in a high proportion of those animals [78]. A better understanding of this pathology in animals may help us to improve their quality of life. Second, animals could be used as models for human medicine and may be useful to improve the diagnoses and treatments of animals and humans. 

Similarly to mTBI, transient ischemic attacks (TIA) can lead to behavioral problems in humans without showing changes in the neurological examination or conventional neuro-imaging techniques (MRI and CT). In fact, from 20 to 50% of TIAs do not show MRI abnormalities [91]. As occurs with TBI, the lack of visible lesions using MRI does not rule out the presence of microscopic and functional damage that can change behavior. Recent studies, where rats were used as models for TIAs, reveal that a similar process could also happen in animals [92]. Transient iatrogenic occlusion of the middle cerebral artery in rats leads to selective neuronal loss and microglial activation, both in striatum [93,94] and the cerebral cortex [92], despite a normal MRI. Loss of normal function of these areas, especially of the cerebral cortex, may lead to long-term behavioral abnormalities. All of these problems are a diagnostic challenge due to the absence of consistent imaging changes or other neurological signs.

## 6. Vomeronasal Organ and Behavioral Problems

Pheromones may be defined as a kind of semiochemical freed from the external surfaces of the body, from where they diffuse into the surrounding environment and change the behavior of the other animals (normally of the same species). Pheromones are perceived by the vomeronasal organ (VNO). The VNO is a part of the accessory olfactory tract, and it is located on each side of the nasal septum [95]. Post-mortem studies in different species show that the VNO can be affected by inflammatory changes [96,97], and these changes may alter the function of any tissue. Thus, inflammatory changes of the VNO may lead to behavioral alterations. Asproni et al. suggest that vomeronasalitis in cats could be related to aggressive behaviors in cats [96]. It could be interesting to investigate the possibility to detect these inflammatory changes in vivo, in order to improve the early detection and the treatment of the problem.

## 7. Cardiology and Behavioral Problems

Cardiac problems may lead to behavioral problems, especially because these problems can decrease the physical activity of the animal. However, it is not common that these problems are referred to a behaviorist, and they usually improve after cardiac treatment.

Notwithstanding, cognitive dysfunction is a problem commonly treated as a behavioral problem that could be in close relation to cardiac problems. Cognitive dysfunction in animals has been typically treated as a mitochondrial malfunction that leads to neuronal oxidative damage, among other factors (environmental factors, neurogenesis, distress, etc.). Nevertheless, this oxidative damage could be due to a reduced or altered blood perfusion of the CNS, and cardiac problems may be the cause of this reduced perfusion. This relation is well established in human medicine [98,99]. Epidemiological studies reveal that 61% of the people with dementia have Alzheimer’s disease (AD), 54% have cardiac problems, and vascular dementia (VaD) and AD are present together in the majority of people [100,101,102,103]. However, although in humans the real prevalence of VaD remains unclear, most of the studies rank VaD as the third most common type of severe dementia in the elderly [103]. Among people with dementia associated with VaD, life expectancy is significantly shortened, when compared to the general population [104]. Although the quality of life (QoL) of these patients has not been widely studied, some authors suggest that decreased QoL in these patients could be significant [99]. The prevalence and physiopathology of this problem has not been studied in animals. Nonetheless, its study could be interesting because animals may be used as animal models for human medicine in order to improve the QoL of animals and persons. In fact, some of the vascular risk factors that have also been linked to dementia in humans (hypertension, obesity, high levels of triglycerides, low levels of High-density lipoprotein (HDL), and diabetes, among others) [105] are present in dogs and cats too. Some of those risk factors are present, even in apparently healthy, middle-aged, and old animals [106]. In light of this evidence, all animals presenting with cognitive dysfunction should be screened for cardiac problems, and proper treatments should be added in case there are any. Thus, the study of VaD should be taken into account in future research in veterinary medicine.

## 8. Conclusions

This revision illustrates the relation between the development of many behavioral problems and different medical conditions. Many of these medical conditions have, as only clinical signs, a behavioral change. Additionally, some of them do not change any hematological or imaging parameter, which makes their diagnoses and their differentiation from a true behavioral problem more difficult. All of this illustrates the importance of taking into account the medical problems in the differential diagnoses of any behavioral problem, and the significance of doing a thorough follow-up in all of the cases.

## Figures and Tables

**Table 1 animals-09-01133-t001:** Patient groups of behavioral problems due to neurological issues regarding abnormalities found in neurological examination and laboratory and imaging work-up.

Group	Abnormalities in Neurological Examination	Abnormalities in Laboratory or Imaging Work-Up	Examples
Group 1	+	+	Brain tumors, brain ischemia, traumatic injuries, etc.
Group 2	+	−	Lysosomal storage diseases, degenerative problems
Group 3	−	+	Tumors of frontal regions of the brain
Group 4	−	−	Idiopathic epilepsy, mild traumatic brain injury, transient ischemia

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
