# Peer review of "A Review of Medical Conditions and Behavioral Problems in Dogs and Cats"

_animals, 2019, doi:10.3390/ani9121133_

Round 1

Reviewer 1 Report

The authors have made all of the suggested changes and this manuscript is, to me, ready for publication. 

I only noticed one very minor thing, where the word 'improve' should be inserted, but that could be done at the proof stage: Line 319, should read: “can help to epileptic dogs to reduce the number of episodes and improve their quality of life”

Author Response

Thank you very much for your help and comments. 

Reviewer 2 Report

My decision is "Accept", after reading the modified version. Best regards,

Author Response

Thank you very much for your help and comments. 

This manuscript is a resubmission of an earlier submission. The following is a list of the peer review reports and author responses from that submission.

Round 1

Reviewer 1 Report

An interesting and extensive piece of work, very useful for clinicians. The addition of some tables, summarizing the most significant behaviour modifications observed with the physical disorders presented in each part of the paper, would have been a very significant improvement for both the comfort on reading and moreover the use of the paper.

Regarding the reference n°92, it would have been more accurate to cite the full work published by Asproni et al. in the journal of feline medicine and surgery, instead of a preliminary report.

Author Response

Dear Reviewer, 

Thank you for your comments and helping. 

Taking into account your comments, we have added a new table (Table 1) and we also have changed the reference nº92 (both in the text and referent list). 

We also think that "Table 1" will help to the readers to better understand this concept. 

Many thanks

Reviewer 2 Report

Reviewers comments for: Medical conditions and behavioral problems in dogs and cats. An article review

Overall comments: I love this paper, it was a joy to read and review and I would like to congratulate the authors on this timely and important review bringing to light an often overlooked fact that health and behaviour are not separate entities but closely intertwined. My comments below are all minor suggestions for improvement only along with a few suggested references the authors may not have come across.

Specific comments:

 Line 54: use of the word ‘bitch’ suggests you are talking only about dogs. Perhaps ‘dam’ would be better?

Line 58-9: “These problems could be divided into four main groups”: perhaps number the following group descriptions to help the reader identify there are four?

Line 68: “most behavioral problems can deteriorate patient’s welfare” this implies the behaviour itself reduces welfare (which it may do of course if the owner then treats the animal worse as a result) but surely the behaviour is most often an indicator of reduced welfare, rather than being causal in reducing the welfare?

Line 73: ‘blind’ should be ‘blindness’

Line 84: remove ‘to lead’

Line 95: I feel like this should be in the lost behaviour category: “decreased interaction with other pets or family members”.

Line 99-100: “However, according to the authors’ experience, pain-related vocalizations, as a main complaint of the owner, are very unlikely” I am unsure what you mean here, perhaps this could be clarified. Do you mean pain-related vocalizations don’t occur often or that owners don’t report them often?

Line 104: typo: ‘experimented’ should be ‘experienced’

Line 113: remove ‘In fact,”

Line 121: doesn’t need to be its own paragraph, it can be the end of the one above. A paragraph should be a collection of sentences, not a single sentence J

Line 142: I would replace ‘with pain’ with ‘experiencing pain’, as pain is an experience, rather than something an animal has or doesn’t have, like they could have fleas for example.  It is the subjective experience of pain after all, that is a key factor in the development of the behavioural issues you describe.

Line 144: ‘short’ should be ‘small’. Perhaps also expand in one or two sentences on the type of relationship this study found? It’s a bit of a hanging point otherwise, i.e. there is a relationship, but the reader doesn’t know what type without going away and reading the other paper.

Line 150: you can remove ‘-being’ and just use ‘human’ (same for line 161 and 175 where you can just say ‘humans’) the same can be done throughout the manuscript.

Line 167: an ‘it’ is needed between ‘because is’

Inappropriate elimination: one thing that has not been considered here is pain from arthritis in elderly cats. In this case, the animal experiences pain upon bending their joints to crouch, which it may choose to avoid by urinating standing up. This means the animal is no longer in pain, but standing to urinate can cause it to ‘miss’ even if it is stood in its tray, which the owner may perceive as a behaviour problem, and the cat itself probably doesn’t like it as then they have a puddle to walk through as they exit their tray, which could cause stress. Appropriate pain medication may reduce or eliminate this issue.

Line 176: replace ‘does’ with ‘the’

Line 177: replace ‘includes’ with ‘can induce’

Line 208-09 not an amendment just a comment: if these authors had seen my cat after she became hypothyroid from radio-iodine treatment they wouldn’t be saying this. She came to an almost complete stop. Slept all the time, was never awake or interacted with the world beyond eating, drinking and elimination and she gained 2kg body weight in 6 months despite being fed the same amount of food that was previously her maintenance weight and only being aged 11 years. She had no quality of life anymore. Treatment with levothyroxine gave her her life back, literally, as she went back to being her normal active and interactive self. I keep thinking I should work with her vet to write it up as a case study as she’s lived healthily for an additional 9 years on levothyroxine since then (and is still going).

Line 226: change “of any aggressive dog” to “of any aggression in dogs”

Line 224: start this with “The”

Line 258: should be ‘especially’

Line 259: I think you mean ‘in the adult’ as opposed to ‘ the adulthood’

Line 264: add the word ‘does’ between ‘this not’

Line 363: this should be the end of the above paragraph as it isn’t a stand-alone paragraph by itself.

Line 379: ‘and’ is a joining word, so the full stop here should be removed to read “referred to a behaviourist, and they usually…”

Line 399: from what you say I think you can recommend here that any animal presenting with cognitive dysfunction should be screened for cardiac problems as well. It may be that treatments for the cardiac issue help to mitigate the cognitive dysfunction, which is an area that should be investigated in future research.

There are a few papers that I expected to see cited in the epilepsy section from the RVC canine epilepsy group but did not. Perhaps consider if these can add anything to it:

PACKER RMA; HOBBS SL; BLACKWELL EJ (2019) Behavioural interventions as an adjunctive treatment for canine epilepsy: a missing part of the epilepsy management toolkit? Frontiers in Veterinary Science. DOI: 10.3389/fvets.2019.00003

WATSON F; RUSBRIDGE C; PACKER RMA; CASEY RA; HEATH S; VOLK HA (2018) A Review of Treatment Options for Behavioural Manifestations of Clinical Anxiety as a Comorbidity in Dogs with Idiopathic Epilepsy. The Veterinary Journal 238, 1-9

WINTER J; PACKER RMA; VOLK HA (2018) A preliminary assessment of cognitive impairments in canine idiopathic epilepsy. Veterinary Record 182633

PACKER RMA; MCGREEVY PD; SALVIN HA; VALENZUELA M; CHAPLIN C; VOLK HA (2018) Cognitive dysfunction in naturally occurring canine idiopathic epilepsy. PLoS ONE 13(2): e0192182

The authorship statement doesn’t fit the journal requirements, perhaps consider adjusting this.

Line 716: I don’t think you can predict this, perhaps reword it? “the dog will never be aggressive again”

Line 735: is this meant to be ‘holy-days’ as in religious ceremonies or ‘holidays’ as in vacations?

Author Response

Dear Reviewer, 

Many thanks for your comments and for helping us to improve the paper. All your comments have been carefully considered and changed when necessary. 

Although all comments and changes have been highlighted in the attached file (word file), I would like to explain here some of the changes: 

Original paper line 58 - now 107:

Line 58-9: “These problems could be divided into four main groups”: perhaps number the following group descriptions to help the reader identify there are four?

We have added numbers to each group. Additionally, we also have added a table (Table 1) in order to summarize the four group and to give some example of each one. 

Original line 68 - now line 117: 

Line 68: “most behavioral problems can deteriorate patient’s welfare” this implies the behaviour itself reduces welfare (which it may do of course if the owner then treats the animal worse as a result) but surely the behaviour is most often an indicator of reduced welfare, rather than being causal in reducing the welfare?

We mean that behavioural problems can directly and indirectly modify animal's welfare because most behavioural problems lead to a stress response that endangers welfare of the animal by itself (for example, most of aggressive-related problems, fear and phobias, etc.) Additionally, at the same time, they can be an indicator of reduced welfare. Finally, welfare also can be affected indirectly, for example because of punishment (if the owners use it as a treatment), or if the owners decide to abandon or euthanise the animal. So, we have included "directly or indirectly" in the text, and if the reviewers consider that we should include that these behaviours can also be indicators of poor-welfare, we will be very happy to change it.

Original line 144 - now line 205

Line 144: ‘short’ should be ‘small’. Perhaps also expand in one or two sentences on the type of relationship this study found? It’s a bit of a hanging point otherwise, i.e. there is a relationship, but the reader doesn’t know what type without going away and reading the other paper.

We have explained in more detail the mentioned study: 

In this study, 20 dogs with noise sensitivity were divided in two groups, 10 dogs that showed noise sensitivity and pain (clinical case group), and 10 that only showed noise sensitivity (control group). In “clinical cases” group, the age of onset of the noise sensitivity was on average 4 years later than “control cases”. Additionally, dogs with pain were more likely to generalize its fear to contextual (environmental) clues and toward other dogs. Finally, “clinical cases” responded well to treatment once the involvement of pain had been identified.  

Line 238: 

Inappropriate elimination: one thing that has not been considered here is pain from arthritis in elderly cats. In this case, the animal experiences pain upon bending their joints to crouch, which it may choose to avoid by urinating standing up. This means the animal is no longer in pain, but standing to urinate can cause it to ‘miss’ even if it is stood in its tray, which the owner may perceive as a behaviour problem, and the cat itself probably doesn’t like it as then they have a puddle to walk through as they exit their tray, which could cause stress. Appropriate pain medication may reduce or eliminate this issue.

We absolutely agree with reviewer comment, so we have added this sentences:

Cats with osteoarthritis, especially in hind limbs, can eliminate out of the litter tray because of pain. These cats tend to avoid bending when urinate and, even when they are standing at the litter tray for eliminating, the owner can find these depositions on the floor, out of the tray. Analgesic drugs and big litter-trays could help to solve this kind of problems.

Original paper line 208-109:

Line 208-09 not an amendment just a comment: if these authors had seen my cat after she became hypothyroid from radio-iodine treatment they wouldn’t be saying this. She came to an almost complete stop. Slept all the time, was never awake or interacted with the world beyond eating, drinking and elimination and she gained 2kg body weight in 6 months despite being fed the same amount of food that was previously her maintenance weight and only being aged 11 years. She had no quality of life anymore. Treatment with levothyroxine gave her her life back, literally, as she went back to being her normal active and interactive self. I keep thinking I should work with her vet to write it up as a case study as she’s lived healthily for an additional 9 years on levothyroxine since then (and is still going).

We absolutely agree with your comment, and we also have some cases of these evidence in our experience. However, we wanted to emphasized the published evidence of this issue.  

Line 389 - 394

There are a few papers that I expected to see cited in the epilepsy section from the RVC canine epilepsy group but did not. Perhaps consider if these can add anything to it...

Thank you for your recommendations, we have include some valuable data of all these studies.

Original line 735 - now line 868:

is this meant to be ‘holy-days’ as in religious ceremonies or ‘holidays’ as in vacations?

Vacations

Many thanks again for all the comments, 

All the very best,